# Experiencing the COVID-19 pandemic as a healthcare provider in rural Dhanbad, India: An interpretative phenomenological analysis

**Laalithya Konduru**[1,2¤a‡]*, **Nishant Das**[3‡], **Gargi Kothari-Speakman**[4¤b], **Ajit Kumar Behura**[3]

**1** Department of Community Medicine, Sri Jagannath Healthcare and Research Center, Dhanbad, Jharkhand, India, **2** College of Medicine and Public Health, Flinders University, Bedford Park, SA, Australia, **3** Department of Humanities and Social Sciences, Indian Institute of Technology (ISM), Dhanbad, Jharkhand, India, **4** Savitri Ghantasala Center for Health Equity, Samanjasa Foundation, Chennai, Tamil Nadu, India

¤a Current address: OHCHR, United Nations, New York, NY, United States of America
¤b Current address: Department of Psychology, University of Nebraska, Lincoln NE, United States of America
‡ LK and ND are joint first authors on this work.
* laalithya@gmail.com

**Data Availability Statement:** Data cannot be shared publicly because the data contains identifiable information, the free availability of

## Abstract

### Background

The COVID-19 pandemic is causing widespread morbidity and mortality. It has led to a myriad of mental health problems, particularly in health care providers (HCPs). To strengthen the fight against COVID-19, it is essential to investigate the mental health challenges being faced by the HCPs, their emotional responses, and coping strategies.

### Objectives

We aimed to explore the lived experiences of frontline HCPs in rural India during the peak of the second wave of the COVID-19 pandemic.

### Methods

Through purposive heterogenous snowball sampling, five HCPs in rural Dhanbad were recruited and one-on-one double-blind unstructured interviews were conducted. The interviews were transcribed and master themes and subthemes were extracted by interpretative phenomenological analysis.

### Results

Six master themes and 23 subthemes were identified. Our findings demonstrate that the participants were under mental duress due to heavy workloads, fear of getting infected and transmitting the infection, urban–rural disparities in access to medical supplies and peer support, and negative social perception of HCPs during the pandemic. Most HCPs have not yet processed the psychological effects of being at the frontlines of the COVID-19 pandemic

which is against the privacy guaranteed to the participants at the time of enrollment in the study. Data are available from the Sri Jagannath Healthcare and Research Center Independent Ethics Committee (contact: The Secretary, Sri Jagannath Healthcare and Research Center - Independent Ethics Committee, Steel gate, Main road, near hero showroom, Dhanbad, Jharkhand - 828127 sweta.sjhrc@gmail.com) for researchers who meet the criteria for access to confidential data.

**Funding:** The publication of this study was funded by a grant from Samanjasa Foundation, Chennai, India (The Babu Rao grant for open access publication of research, 2022).

**Competing interests:** The authors have declared that no competing interests exist.

in a resource-poor setting; however, spirituality seems to be an important coping mechanism that helps them get through the day.

## Conclusions

This study is unique in the sense that not many studies have been conducted to evaluate the psychological issues of Indian HCPs during this pandemic. Much less is known about the mental health of HCPs in rural settings. Moreover, novel findings such as negative social perception of HCPs during the pandemic and HCPs resorting to spirituality as a coping strategy against stress, open a plethora of research opportunities wherein the results of this qualitative study, along with the existing literature and findings of future quantitative studies, can establish better understanding of the impact of the pandemic on HCPs.

## 1. Introduction

The coronavirus disease 2019 (COVID-19), as on 8th August 2022, has affected 590,569,394 people and claimed 6,439,499 lives worldwide [1]. This disease—an acute respiratory disease caused by exposure to a novel coronavirus [2] that was first detected in China—has taken the form of a pandemic and has become the most serious menace to public health since the H1N1 influenza outbreak of 1918 [3]. In addition to the threat to physical wellbeing, the COVID-19 pandemic has threatened the mental wellbeing of people at large [3, 4], and has caused widespread and severe mental health problems [5]. A rise in the levels of stress and anxiety has been observed in the general population [6]. In particular, healthcare providers (HCPs) have experienced heightened levels of stress [5]. A study of 1563 HCPs by Ho et al. [7] showed that HCPs are experiencing anxiety (44.7%), depression (50.7%), and sleep disturbances (36.1%). The world Health Organization (WHO) anticipates that HCPs may develop posttraumatic stress disorder (PTSD) after the cessation of the pandemic [8]. There could be a variety of reasons behind the increased stress, ranging from increased workload due to higher demand for healthcare services to stressors that are social in nature; it is extremely important to identify and mitigate the stressors in order to boost the morale of HCPs. A study [9] showed that during the pandemic, a majority of research papers on topics related to COVID-19 have come from China (39%) and the United States (28%), while research publications from developing countries have been "relatively absent". Naturally, studies on the mental health of HCPs during the COVID-19 pandemic have also been mostly conducted in China and the United States. At a time when a pandemic continues to spread and wreak havoc on public health, the mental wellbeing of HCPs is of utmost significance as they are frontline warriors who can minimize the damage caused by the pandemic. To date, very few studies have been conducted to evaluate the psychological issues of Indian HCPs [10]. Much less is known about the psychological issues of HCPs in the rural settings of India, and even less is known about their psychological issues during this pandemic.

Since the start of the pandemic, we have collaborated with the public health authorities of Dhanbad, India to investigate whether the frontline workers in different settings of the district could be supported in any way in their fight against COVID-19; as part of this endeavour, this study aimed to investigate the lived experiences of the participants and at the same time capture any generalisation, if discernible. Due to the enormous stress experienced by the HCPs during the COVID-19 pandemic [4, 11], we aimed to use the results of the current study and other completed and ongoing studies resulting from our collaboration with the public health authorities of Dhanbad, India, to subsequently design an intervention to mitigate stress and

tailor it in such a way as to maximise its use by the frontline workers of the COVID-19 pandemic in Dhanbad, India. We believe that it is vital to take notice of individual accounts in order to determine effective interventions tailored to the population of interest; therefore, a qualitative method was considered to be best suited to achieve this goal. Among the main methods of qualitative research (phenomenology, grounded theory, discourse analysis and narrative analysis), the phenomenological approach was preferred as the study is concerned with understanding the experience of a particular phenomenon (delivering healthcare during the COVID-19 outbreak). As the study aims to keep one eye on the individual accounts, while keeping another eye on any possible generalisation—which can be achieved by an analysis of the convergence and divergence of individual accounts—within the phenomenological approaches, interpretative phenomenological analysis (IPA) was chosen as it shows both convergence and divergence between participants, while retaining their individuality [12]. Nomothetic methods were rejected as they are probabilistic, providing group averages wherein individual experience is lost [13]. Thus, using IPA, this study aims to qualitatively examine the lived experiences of HCPs working in rural settings in India, and elucidate the challenges they are confronted with while discharging their duties during the COVID-19 pandemic and the coping mechanisms they have developed to deal with the same.

## 2. Methods

The lived experiences of HCPs were explored using IPA. IPA emphasizes the explanation of a phenomenon as experienced by people, and their nomothetic interpretation of the experience, which is then cogitatively interpreted by the researchers [14]. Thus, it is a double hermeneutic method [14] in which the respondents interpret the meaning of their own experiences, while the researchers interpret these interpretations. A sample size of 3–12 is acceptable for IPA [15].

### 2.1 Ethical considerations

The study was conducted in compliance with the Declaration of Helsinki and the national guidelines. Prior approval was taken for all the procedures from Sri Jagannath Healthcare and Research Center Independent Ethics Committee (No: SJHRC/21/APR/16).

### 2.2 Participants

The inclusion criteria were (1) age >18 years, (2) involved in frontline healthcare services during the COVID-19 outbreak, (3) consenting to participate in the research and to publish their anonymized interview excerpts. Through purposive sampling, one HCP in rural Dhanbad, India, identified via referral from HCPs known to the authors, was informed about the study, and written informed consent to participate was obtained. The participant was then asked to identify another suitable HCP working in rural Dhanbad, India, but belonging to a different healthcare profession to their own, for inclusion in the study. In this way, through purposive heterogenous snowball sampling, five HCPs in rural Dhanbad, India, were informed about the study, and all of them gave written informed consent to participate. The participants were chosen to represent the breadth of experiences of different healthcare professions, at different stages of training.

### 2.3 Setting and data acquisition

The study is set in the rural areas of Dhanbad, India. Dhanbad is a district in the Indian state of Jharkhand. According to the 2011 census, Dhanbad district has a population of 2,684,487 individuals, 53% of whom are male; the literacy rate is 75.71%, the poverty rate is 26.76%, and 80.07% of the population is Hindu [16]. Jharkhand has a doctor–patient ratio of 1:8200 [17].

Data acquisition for this study was completed during the second wave of the COVID-19 pandemic in India, between 27th May, 2021 and 4th June, 2021. Each participant was assigned a pseudonym at enrolment, and they were instructed to use only their pseudonyms in all their interactions with the researchers. The researchers also used pseudonyms. Thus, the interviews were conducted in a double-blind manner. Unstructured interviews were conducted over phone calls; a list of questions was not prepared. Unstructured interviews were preferred over semi-structured ones as they offer an opportunity to capture the experience of the interviewee in the most concrete form possible [18, 19], which is central to the phenomenological underpinnings of IPA. In addition, the unstructured nature of the interviews allows exploration of unanticipated paths which reflects the inductive epistemological position of IPA [20]. Moreover, the unstructured method also helps reduce the status difference between the interviewer and the interviewee [19]. Additionally, in unstructured interviews, the interviewee is central to the process and commands most of the control of the process and content [21, 22]. Therefore, we opined that the HCPs, who were under enormous stress during the pandemic [4, 11], would feel much more at ease during the interviews, and the interview process might even provide a therapeutic benefit by giving them an opportunity for emotional catharsis [18]; thereby aligning with the goals of our collaboration with the public health authorities of Dhanbad, India. An opening question—"What is it like to be on the frontlines of the COVID-19 pandemic in a rural setting like Dhanbad?"—was used. Audio was recorded on an iPhone 10, and transferred to a password protected google drive. The interviews were in English, took 60–90 minutes, and were completed over 1–2 phone calls—some interviews were interrupted as the participants had to attend to patients. Nobody other than the researcher and participant was present during the interview. We adopted an individual-oriented approach to data saturation [23] and defined data saturation as the point at which the participant began repeating the same information without much change in the non-verbal cues associated with that information. Data was acquired until saturation was achieved.

## 2.4 Data analysis

The interviews were transcribed verbatim, and in order to get an overall impression of the contents, a simultaneous reading of the transcripts was done alongside listening to the corresponding audio recordings. Then the first authors contemplated on their predispositions and personal experiences associated with the study question and were debriefed by the third author. Subsequently, they individually re-read the transcripts several times, following which, all statements concerning each participants' experiences of or meaning ascribed to the phenomenon of interest (experiences while working as an HCP during the COVID-19 pandemic in rural India) were captured into emergent themes. Interrelated themes were grouped together into subsidiary themes, capturing the participants' descriptions and the authors' interpretations of the descriptions. Subsidiary themes were then condensed into principal themes, capturing the gist of the participants' lived experiences. Finally, the first authors individually extracted the principal themes from all transcripts into a set of master themes and subthemes. The second author prepared a consolidated list of master themes and subthemes using the lists of the first authors, and reconciled differences between the individual lists by facilitating a discussion between them.

## 3. Results

### 3.1 Participant characteristics

There were five participants (two male and three female) in this study. They were adherents of the three major religions of India. The median age of the participants was 43 years, with a range of 40–56 years. The median number of years of experience after graduation was 18

years, with a range of 7–35 years. Doctors, paramedics, nurses, and midwives were represented in the sample. The demographic characteristics of the participants are presented in Table 1.

## 3.2 Investigation of individual themes

All significant utterances of each participant that underpinned the analysis are presented in the S1 Appendix. The subsequent subsections briefly present the individual themes of each participant.

**3.2.1 Varun.** Varun provides the perspective of a senior critical care specialist. He has made sense of the situation as being in a war and he sees himself as a soldier in this war. When talking about the pandemic, he repeatedly mentioned the risk to his marital partner, while alluding to his own risk of contracting COVID-19 only once, that too when talking about the risk to HCPs in general. He is not satisfied with the standard of care he is providing because the hospital he works for has restricted cardiopulmonary resuscitation in COVID-positive patients given the virus-aerosolization (generating virus-containing fine droplets that can remain suspended in air for long periods) risk, and because he is unable to explain the deaths of his patients. He is afraid of missing some potential treatment, so he reads recent research and treatment guidelines and follows them diligently. Unless he has tried every measure to save his patients, he feels morally conflicted. He feels that his exposure to COVID-19 is a given and is willing to go to great lengths to save others, even at a great personal cost. A sense of frustration at the uncertainty of the pandemic, high cost of treatment, and the careless attitude of the public is palpable from his account of the pandemic and his non-verbal cues. His attitude towards his work and general outlook during the pandemic appears to be consistent with that of a soldier at war, willing to make the supreme sacrifice.

**3.2.2 Vayu.** Vayu provides the perspective of a first responder. He repeatedly mentioned engaging in frontline COVID-19 work due to financial constraints. If he did not have a family to provide for, he would not work as a frontline HCP as he is afraid of contracting COVID-19 given his diabetic status; thus, he has placed the needs of his family above his own needs. At one instance, he narrated an incident where he did not remove an abusive individual from his ambulance because this individual's mother—the patient—required oxygen support. He felt that the patient would not be comfortable without her son; he displayed empathy towards his patient and put her needs above his own need to remove himself from an abusive situation. Although he has made sense of his engaging in work during the pandemic as a compulsion due to financial needs, he also seems to derive a sense of self from his work as a paramedic; he lost his sense of self during the pandemic due to witnessing the deaths of patients owing to a shortage of beds and oxygen and his helplessness in the face of COVID-19.

**3.2.3 Sani.** Sani provides the perspective of a doctor who is not trained in managing acute respiratory diseases but had to treat severe cases of COVID-19 during the second wave of the pandemic in India. She repeatedly mentioned her lack of training throughout her account. She feels she is not the right person to be treating severe cases of COVID-19 and is questioning her

**Table 1. Participant demographic characteristics.**

| Pseudonyms | Sex | Age (Years) | Number of years after graduation | Occupation | Religion |
|---|---|---|---|---|---|
| Varun | Male | 56 | 35 | Anaesthesiologist | Christian |
| Vayu | Male | 40 | 12 | Paramedic | Christian |
| Sani | Female | 43 | 07 | General Physician | Hindu |
| Bhumi | Female | 47 | 25 | Nurse | Muslim |
| Agni | Female | 40 | 18 | Midwife | Hindu |

decision to volunteer for frontline COVID-19 work. Deaths of patients due to drug shortages adversely affected her mental health; in a way she feels responsible for those deaths. She repeatedly mentioned God during the interview and is drawing strength from her spiritual practices to deal with the pandemic and with her doubts about her ability to treat severe cases of COVID-19. She has made sense of her experience of the pandemic as someone who made an ill-informed wrong choice, albeit an altruistic one, and is doing her best to avoid its negative consequences on the wellbeing of others.

**3.2.4 Bhumi.** Bhumi provides the perspective of a senior nurse. She repeatedly mentioned a lack of financial reward for her work during the pandemic. She is dissatisfied with the compensation she is receiving for her work, but she feels a strong sense of belonging with the community of HCPs, which motivates her to continue working. Lack of acknowledgement from patients for her contribution to their recovery has adversely affected her self-worth. She has not yet made sense of her experience of the pandemic, and is mechanically going about her life and work. She feels that if she reflects on her experiences and tries to make sense of them, she will not be able to cope; she has decided to put off this exercise until the end of the pandemic and concentrate on completing the work that is assigned to her. Interestingly, although she feels close to her family, she was the only participant who did not mention family at all during the interview, other than a passing reference to her marital partner.

**3.2.5 Agni.** Agni provides the perspective of an allied health worker who is not trained to manage lung diseases but had to manage severe cases of COVID-19 during the second wave of the pandemic in India. Unlike Sani, she did not question her decision to volunteer. In fact, she had a positive attitude towards her decision to volunteer and regardless of training, she did not feel someone else could have done a better job than her. She felt a strong sense of responsibility towards her patients, from which she drew motivation to continue working during the pandemic. The uncertainty surrounding COVID-19 was a recurrent motif in her narrative; it threatened her sense of safety. A threatened sense of safety severely impacted her quality of life. The meaning she attached to the pandemic was that of a poorly-understood phenomenon that is to be feared. She felt disconnected from her family as she felt they do not share her experience of the pandemic. She felt only her peers could understand her perspective, but due to stigma and fear of appearing weak in front of her peers, she did not discuss her fears with them.

## 3.3 Investigation of the master themes

Six master themes were identified (Table 2). The important deductions from all the subthemes along with the prototypical edited excerpts that capture the essence of each subtheme are elaborated in the subsequent subsections.

### 3.3.1 Mental challenges in the discharge of duty.

*a. Fear of transmission.* All participants expressed a fear of contracting COVID-19.

> **Agni**: "I follow all the guidelines for COVID safety. But you never know; anything can happen."

They also expressed a fear of transmitting COVID-19, and the fear of infecting their family members seemed to be predominant. They remained constantly worried about protecting their family.

> **Sani**: "I always keep thinking, what if I bring it home from the hospital."

**Table 2. Outline of master themes and subthemes.**

| S. No | Master theme | Subtheme |
|---|---|---|
| 1 | Mental challenges in the discharge of duty | Fear of transmission |
| | | Fear of inadequate training |
| | | Society's perception of profiteering |
| 2 | Infrastructural constraints | Lack of supplies |
| | | Insufficient staffing |
| 3 | Self-perception of inadequate care provision | Lack of peer support |
| | | Self-perceived inferior quality of care |
| 4 | Emotional responses | Helplessness |
| | | Hopelessness |
| | | Anger on people who do not use masks |
| | | Dissatisfaction upon not getting recognition |
| | | Satisfaction on live discharges |
| | | Sense of responsibility |
| 5 | Coping mechanisms | Psyching oneself up |
| | | Deriving motivation from a sense of responsibility |
| | | Spirituality |
| | | Being on autopilot |
| | | Not coping |
| 6 | Ideas on how to handle the pandemic better | Counselling |
| | | Rest |
| | | Increase in workforce |
| | | Increase in remuneration |
| | | Adequate supply of PPE |

Some of the participants took extraordinary precautions to avoid transmitting the virus to their loved ones.

> **Varun**: "We only have one bedroom, so to avoid passing the infection to my husband, I have been sleeping in the kitchen since the pandemic started."

Apparently, the fear of endangering their loved ones ensured a stricter observation of the safety guidelines. Their fear of carrying the virus home probably stemmed from their exposure to patients with COVID-19 throughout the day. Possibly, factors like the high virulence and infectivity of the virus were at the root of their apprehensions. Participants with greater vulnerability to COVID-19, such as those with comorbid conditions, feared for their own safety.

> **Vayu**: "Every time I get a call, I pray that it is not shortness of breath. You see, I have diabetes, which makes me really vulnerable to COVID."

Reports of HCPs contracting and dying of COVID-19 might have contributed to this fear. COVID-19 precautionary measures like washing hands regularly, wearing a mask, observing social distance etc., were followed diligently by the participants to avoid transmitting the virus.

> **Bhumi**: "I think I have developed OCD; I wash my hands too many times. I know all my patients have COVID, but I do not want to cross-infect anyone."

*b. Fear of inadequate training.* The Governments have been utilizing all the human-resources at their disposal, including doctors regardless of specialization, paramedics, nurses, and midwives, in the management of the COVID-19 pandemic. Not all the HCPs had prior experience or training in treating patients suffering from severe respiratory distress and/or fatal viral infections. As a result, the participants seemed to have fears about their ability to treat the patients adequately. Participants admitted to having insufficient understanding of the disease.

**Agni**: "Sometimes I feel as if I am as clueless as my patients' family"

Some participants felt that specialists could have done a better job than them.

**Sani**: "There are no specialists here, I have to be everything that my patients need me to be. But I cannot help but think that maybe because I volunteered, my patients are getting a poorer deal. If I hadn't volunteered, the government would have sent a specialist, and they would have done a much better job. They are trained for these things, I am not."

The HCPs also felt they were not trained to handle a large caseload.

**Varun**: "All these years of practice, and I still don't feel fully ready. There are just too many patients, and I am not trained to handle this kind of caseload."

The speculations surrounding the definitive cure of COVID-19 and the absence of specific training for treating patients with COVID-19 might have added to this trepidation. Consequently, a sense of guilt was palpable in some of the responses, for not being able to do everything for their patients that a well-trained specialist could have done.

*c. Society's perception.* The participants felt vilified by the society for some reason or other.

**Sani**: "When I volunteered to treat, a friend said, oh so you've joined the goldrush!"

Most HCPs experienced being perceived as exploiters by the society, probably due to the high charges of beds, drugs, and overall treatment cost.

**Varun**: "The people were generally very grateful. But when I mentioned the cost of the drugs, they saw me like a thief."

**Bhumi**: "The patients' families are thinking we are looting them. Fact is, my salary hasn't increased in two years. Even with such an increased patient load, I am drawing the same salary, but I am doing my job with the same commitment without complaining. I don't understand why I'm the enemy."

Some of them experienced hostility from the family members of the patients for the delays in bed allocation.

**Vayu**: "I was transporting a patient to the hospital. As there were no beds, I was asked to wait outside the hospital gates, along with many other ambulances. One by one the ambulances were asked to come in as and when beds were available. Our turn came after 4 hours. The patient's son started shouting at me after sometime. He said this delay was just drama so that the hospital can ask for more money and we are putting patients at risk for money

by doing this drama. He started non-stop abusing me; I did not throw him out only because it was a life and death question for the patient."

Some respondents felt that the expression of gratitude by the society was only pretence.

**Agni**: "Outwardly, they're all acting like they're grateful to us, but inside they all think we are looting them."

### 3.3.2 Infrastructural constraints.

*a. Lack of supplies*. Participants admitted feeling traumatized by the deaths of patients who were waiting outside the hospitals due to a combination of shortage of oxygen and shortage of beds in the hospitals.

**Vayu**: "We get a small portable cylinder that lasts maybe 1 hour when we go to see a patient. After that we have to shift to the ambulance, and the cylinder in the ambulance lasts maybe 6 hours. But we are not allowed to shift the patient to the ambulance unless the family has a bed confirmed at a hospital. But because of bed shortage, some people died after the portable cylinder ran out. When I have to wait for beds outside the hospital, lucky patients got it before the cylinder in the ambulance ran out, but there were unlucky ones whose turn did not come. Their deaths haunt me all the time."

At times, due to shortage of drugs, the HCPs had to further triage and allocate drugs to their patients, which they felt meant choosing who would live and who would die. They felt uncomfortable dealing with such a moral dilemma.

**Sani**: "We don't have enough drugs; we have to ration them. I feel like I am playing God, choosing who gets the drugs and who doesn't. It is not right."

A rural–urban disparity in the distribution of resources was observed as some HCPs highlighted that their demands for essential supplies were met only after those of the urban hospitals were met, and that they got delayed deliveries. They complained of being neglected by the government. It seemed that this perceived discrimination had a negative impact on the self-esteem of the HCPs deployed in rural areas.

**Bhumi**: "We get our supplies of oxygen and PPE and drugs and everything only after all the metropolitan guys get their share. Whatever is left is very little, and that has to be shared among many hospitals in the rural places. Our needs are underestimated, and if we ask for more, we never get it on time. We are like Cinderella and the hospitals in the metropolitan areas are our stepsisters and the government is our stepmother."

*b. Insufficient staffing*. All respondents highlighted the issue of inadequate human resource. They expressed the desire to be able to share their workload.

**Sani**: "There is nobody to lean on. Just me and a nurse and a midwife. And so many patients!"

**Varun**: "I wish there was someone I can share my caseload with. But no such luck. I have to deal with this alone."

The unavailability of their replacement meant that they had to work with minimum breaks, enduring physical strain that affected their bodies.

> **Vayu**: "My break is while driving, and the driver takes a break when I assess the patient. We have no dedicated breaks otherwise."

> **Bhumi**: "I have no rest. My whole body aches, but I have to go on because there is nobody else to do my job."

It also meant that they could not avail leaves to recover.

> **Agni**: "I want to take some time to recover, but my leave was not approved because there is no replacement."

### 3.3.3 Self-perception of inadequate care provision.

*a. Lack of peer support*. It was observed that due to the workload, the HCPs had to work with minimum or no breaks. They were not able take even the food, drink, and restroom breaks. Sometimes they worked for 30 hours before another colleague took over. While enduring these hardships, they witnessed the deaths of their patients, leading to considerable impact on their emotional state.

> **Varun**: "Day before yesterday was so bad that I couldn't even take a break to go to the bathroom. I had breakfast before coming to work, and then I had one glass of tea at 8 'o clock at night, then went back to the ICU. No water, no toilet. I worked all night as well, and I lost 4 patients. It was the worst day of my life. There was nobody to relive me, I worked alone for 30 hours before a relieving doctor came. The thought that maybe I lost those patients because I was so tired that I didn't do everything right has come to my mind several times."

HCPs felt the need for peer support whenever they had doubts about the treatment or had to perform procedures that they were not routine for them. They felt helpless on such occasions and tried contacting their friends for help.

> **Sani**: "If I have a doubt, there is nobody to ask. I have to phone a friend, and if they don't pick up, try another one. If they tell me to do something which I haven't done in a long time, then that's too bad, because only I have to do it, no referring to someone else. Because there is no supervision at such times, I am always feeling I am not doing the best job. This is not the best level of care that patients deserve."

*b. Self-perceived inferior quality of care*. The safety constraints put in place during the pandemic by some healthcare institutions, such as not allowing HCPs to perform cardiopulmonary resuscitation (CPR), had curtailed the quality of care the participants were used to providing, leading to dissatisfaction. Some of them admitted to being morally conflicted.

> **Varun**: "We are trained to do everything we can for a patient, but our hospital has decided not to do CPR on patients because of the aerosolization risk. I cannot in good conscience say that I have done everything possible."

The highly contagious nature of the virus had effectively removed any human touch from healthcare, which according to the participants, was an essential component of care. The

protective gear had apparently taken away the human side of caregiving and made their job mechanical. They seemed visibly discontent with the fact that they were made to provide what they deemed as incomplete treatment.

> **Bhumi**: "We cannot touch the patients without wearing full PPE. But touch is a very big part of healing, it shows the patient that we care. And if I smile also the patient does not know. Now everything feels mechanical. It is not the way to care for the patients."

Some participants expressed self-doubt and blamed themselves for the deaths of the patients under their watch.

> **Sani**: "When I volunteered, I thought I will just be assessing patients and sending severe ones to the hospital. But there are no beds, and I am treating the severe ones also. How is my care going to be good? The patients I have lost, maybe if they went to a well-trained specialist at a well-equipped hospital, they might have survived, who knows?"

### 3.3.4 Emotional responses.

*a. Helplessness.* Most HCPs felt helpless for some reason or other. In the absence of a definitive treatment, HCPs did their best to keep themselves up to date with the latest research and yet, they lost many patients. Naturally, they felt helpless. A sense of self-reproach, for not being able to do anything, was evident.

> **Varun**: "I follow all the treatment guidelines, I read up on research, I do everything to the letter, but patients still die. I cannot figure out why, and I cannot live with that. I need to know what I could have done better. I need to know what I got wrong, why did my patients die."

Seeing patients die before even reaching the hospital made the HCPs feel helpless, more so because the families of patients pinned all their hopes on them.

> **Vayu**: "Sometimes when I reach a patient's home, they are already in bad shape, I know they may not even reach the hospital alive. The family ask me to do something, but I am so helpless, there is nothing to do."

They also felt helpless at being unable to allow the family members to accompany their loved ones in their last moments.

> **Agni**: "I have seen so many people dying alone, without any loved one near them in their last moments. Their families plead with us to let them be with the patient at the end. The helplessness that I feel at these times, I have never felt it before in my life."

*b. Hopelessness.* The pandemic had been continuing for so long as to seem permanent, leading to hopelessness among the HCPs who have been fighting it out since day one.

> **Sani**: "Will this end? I just cannot see an end to all this suffering."

Witnessing the deaths of patients under their watch during this long-lasting pandemic had taken a heavy toll on the self-worth of the participants.

**Vayu**: "So many people have died on my watch that now I feel everything is pointless, my work, my life, everything."

*c. Anger.* There was palpable anger against people who do not follow COVID-19 safety precautions. The participants felt that such people were putting the lives of others at risk.

**Varun**: "When I see anyone venturing out without a mask, I give them a piece of my mind. We are risking our lives, and they're putting us at more risk unnecessarily."

**Sani**: "Don't they see how many people are dying? I really don't understand why they are so stubborn to cost more lives. They should be jailed."

Some participants confronted such people, handed them masks, and compelled them to put them on.

**Vayu**: "I hand out masks to those who don't wear them, and I don't get out of their way till they wear it. Some people have been annoyed by it, but I don't care."

**Bhumi**: "It makes me so mad. I have stopped them on the road and handed them masks. My husband thinks I am going to irritate someone so much that they might hit me, he always asks me to stop. But seeing so many deaths, I know I will never stop doing this."

Some HCPs even called the police on them.

**Agni**: "I have called the police on such people. They should be booked under the pandemic act."

It appears that the HCPs saw people who did not follow COVID-19 safety precautions as the reason behind the pandemic not abating and therefore, the reason behind their stress.

*d. Dissatisfaction upon not getting recognition.* One of the participants perceived that HCPs other than doctors were not getting due recognition for their efforts. She complained that the hard work put in by the nurses was not acknowledged by the patients after their recovery. Seemingly, this had adversely affected her self-worth.

**Bhumi**: "The patients don't thank us when they get better. They thank only the doctor. We also do a lot of hard work. Why don't we get recognized as much? In fact, we spend more time with the patients and their family members than the doctors, then they should be thanking us more, instead they thank the doctor and go away. Is my work so worthless? I feel worthless."

*e. Satisfaction on live discharges.* It was highly gratifying for the HCPs to see the sight of a successfully treated patient; more so because they had seen so many deaths under their watch during the pandemic. They derived a sense of accomplishment and reward and encouragement upon a successful live discharge.

**Varun**: "There is nothing better than watching a patient walk out of the hospital."

**Vayu**: "When I reach a patient on time, or when they reach the hospital safely and get a bed, I feel very happy that they get a real chance to fight. I feel I have done my job."

**Sani**: "All the doubts and frustration are gone when I am signing the discharge papers and the patient goes home to their family. It gives me a sense of fulfilment."

*f. Sense of responsibility*. Some respondents admitted that the pandemic had been long and demoralizing, and that the thought of quitting had crossed their minds. However, out of a sense of responsibility towards their patients, they did not quit.

**Sani**: "I have thought of quitting so many times. But it would be irresponsible of me to quit now, when there is so much need."

The respondents took pride in making a difference to the patients' lives and displayed a realization of the importance of their work.

**Agni**: "This pandemic has gone on for too long, work has become boring. But these patients are my responsibility. The only thing that keeps things sane is the fact that I know that I am making a difference to them."

### 3.3.5 Coping mechanisms.

*a. Internal dialogue*. Positive self-talk was an effective method of self-motivation for some HCPs. They admitted that after an unnerving day's work, they encouraged themselves through internal dialogue, which mentally prepared them for the next day.

**Varun**: "I take a long bath when I go home and during the bath, I talk to myself, I tell myself that this war is not over, you may have lost today but tomorrow will be yours, now go and sleep and get ready for tomorrow. It may sound crazy, but it works for me."

*b. Deriving motivation from a sense of responsibility*. Some HCPs expressed that a sense of responsibility towards their family was a driving factor for them in their incessant toil.

**Vayu**: "I don't know. I wish to hug my wife, but because I don't want to infect her, I sleep outside. I just see her and my family safe in the house and tell myself that if they have to be safe, I have to be outside and I have to be alive. I need to be there for them, and that is the thought that balances me."

Meanwhile, some HCPs derived strength from a sense of responsibility towards their patients. They felt that they needed to be strong, despite their mental woes, because they needed to be there for their patients. It reflects a strong sense of duty and obligation helping them draw motivation during difficult times.

**Agni**: "The patients need us right now; we have to be strong. But the truth is, I am scared. I don't know what will happen. I am unable to sleep properly. I worry a lot, but when it is time to go for duty, I just shut off my mind and go because my patients need me."

*c. Spirituality*. HCPs confessed that their fears and worries could only be understood by their peers as, unlike their friends and family, they were the ones risking their lives. But they could not comfort each other due to their busy schedules and also due to the stigma about mental health issues prevalent in the healthcare community. Apparently, the fear of

appearing weak subjugated their urge to share their stress with each other, thereby making spirituality their only resort.

**Agni**: "I talk to my family, but end of the day, no matter how they try to help me, no matter what they say, it is me who is risking my life. Nobody can understand my fear and what I am going through. And my peers who will understand, they are too busy and also if I say I am afraid, even if they themselves are afraid, they will see me as weak and unworthy of working in healthcare. So, I don't talk to anyone. Instead, I pray."

HCPs took refuge in faith. Praying and reading religious texts kept their hopes of return to normalcy alive, and inspired them to perform their duties.

**Sani**: "Every day at prayer time, we have the habit of reading a verse from the Ramayana. When I read the verse, I remember that despite so many struggles if Prabhu Ram can be so calm and perform his duties, I too must learn from him and do my duty. Only then he will bless me."

**Agni**: "God is the only one who will listen and not judge me for my fears. I just pray to him that this too should pass and I should be alive by the time things return to normal."

**Varun**: "My church has organized mass through zoom. It has been the best thing. It gives me a sense of hope and community and helps me take my mind off things."

*d. Being on autopilot.* Most HCPs, notwithstanding their fears and worries, went about their job without thinking about anything else, as if being on autopilot.

**Bhumi**: "I am not coping. Right now, there are so many patients that I am not getting time to process what is going on. The day this ends, I am sure I am going to have a nervous breakdown."

Evidently, the workload did not allow them any time to ponder over any of their concerns or the ways to handle them during their shift hours.

**Varun**: "While working, there is no time to think, it is like I am on autopilot."

While some HCPs, after shutting their minds off from everything else during their duty hours, found ways to cope after work, others continued to drift along their daily routine in autopilot mode without any coping strategy. They seemed to be in a very fragile mental state as they admitted not finding any time to even understand what was going on, let aside coping with them.

**3.3.6 Ideas on how to handle the pandemic better.**

*a. Counselling.* The participants expressed the need for counselling for their mental wellbeing. They admitted that even though seeking help for mental health was stigmatized in the healthcare community, the pandemic showed them that they were not immune to human mental frailties and that they too require mental support.

**Varun**: "We all need support. Mental wellbeing, especially among doctors, is taboo. We see it as a sign of weakness. But if anything, this pandemic has taught us, it is that even the best

of us needs a helping hand now and then. Counselling should be made accessible to all healthcare workers."

*b. Rest.* Most HCPs were rest-deprived and overworked due their extensively stretched schedules.

> **Varun**: "We need time off to spend away from hospitals and patients. We are already burnt out, if this continues, many doctors will simply leave medicine."

They expressed the need for breaks to spend time away from the patients and hospital. It seems their nonstop routine throughout the pandemic had led to physical as well as mental fatigue.

> **Agni**: "I don't need to be treated like a hero, just want to be treated like a human being. I also have a family; I also need rest."

*c. Increase in workforce.* All HCPs highlighted the shortage of staff and suggested different ways to increase the capacity in order for them to get a reasonable amount of rest.

> **Varun**: "They should hire more doctors so that we can work reasonable shifts and get enough rest."

> **Agni**: "I want a break, for that rural hospitals should hire more nurses."

A major rural–urban disparity in staff capacity came to light as the HCPs deployed in rural regions pointed out the yawning gap between the demand and availability of healthcare workforce in rural areas, possibly because the capacity increase was mostly limited to the cities.

> **Vayu**: "Whatever staff increase has happened; it has all been in the cities. We have been forgotten, but sadly, COVID hasn't forgotten us."

To overcome this gap, the participants suggested that foreign trained doctors could be allowed to work in the rural and suburban regions without having to pass the exam of the Medical Council of India (MCI).

> **Sani**: "Why can't they give special permission to overseas trained doctors to practice without the MCI exam? They can be supervised by an Indian licensed doctor even. And make a rule that they can work without the MCI exam only in rural areas for the duration of the pandemic. If students can be allowed, why not them? Our staffing problems will be gone."

The participants endorsed utilizing the medical and nursing students but said the government must devise ways to bring more HCPs to rural areas.

> **Bhumi**: "The idea to involve medical and nursing students in the care of COVID patients is a good idea. But if you see, there are very few colleges outside cities. We continue to be short staffed. The government needs to do something to bring more doctors and nurses to the rural areas."

*d. Increase in remuneration.* Some of the participants wanted recognition from their management for their work and believed that they deserved a bonus and an increase in salary for

putting their lives on the line. It seems that they felt undercompensated. They also proposed an increase in pay as a means to incentivize working in rural areas.

**Bhumi**: "Maybe if we can get paid more, we will also be happy and more people will also be willing to work in rural areas."

**Agni**: "I want better pay. Our management must recognize our efforts and give us a bonus. We are after all risking our lives, we should get a bonus and salary hike. I think we earned it."

e. *Adequate supply of Personal Protective Equipment (PPE)*. Some participants highlighted the need to ensure adequate supply of PPE as it helped them feel safe.

**Agni**: "I just want to be safe, for that we should get enough PPE. I should not think about when the next supply of PPE will come and should I throw this mask or not and get a new one, what happens if we run out. Those are unnecessary distractions from our duties and they put so much mental pressure on us. It should just be taken care of."

## 4. Discussion

The IPA method applied by this study provided insights into the lived experiences, mental health status, and coping strategies of the HCPs in rural Dhanbad, India, as well as their ideas for improvement. Previous studies have found that HCPs have been experiencing increased psychological distress during the pandemic [4, 13, 24]. A study conducted on 1,257 HCPs in China, showed that 71.5% of HCPs sustained psychological distress, 44.6% had anxiety, and 50.4% had depression [25]. This study also showed the presence of a gamut of mental health issues among the HCPs working in a rural setting in India during the second wave of the COVID-19 pandemic. Major causes of distress of the respondents were: fear of infection and infecting family members, workload, diminishing workforce, insufficient PPE, social perception, isolation, and uncertainty of the future. These findings were in consonance with the findings of other studies on mental health of HCPs during the COVID-19 pandemic [4, 25].

As the number of HCPs gradually reduced due to illness, and the patient count kept increasing [26], the patient volume became disproportionately higher than the HCP capacity during the pandemic. Consequently, the HCP workforce became overburdened [27] and the HCPs have been stretching their work shifts beyond the usual hours; as a result, HCPs have been unable to take time away from work to rejuvenate. In our study, we saw that being unable to unwind was a significant cause of distress. All participants mentioned that they needed rest and their managements must expand the workforce so that they can work more reasonable hours. It can also be said that the excessive workload prevented some of the participants from processing their emotions, and they have been putting off dealing with the psychological fallout of being on the frontlines of the COVID-19 pandemic for a prolonged period of time in order to serve their patients.

There have been reports that HCPs have been resigning their jobs during the pandemic, and the main reasons cited for the resignations included fatigue, not availing leave since the start of the pandemic, migration to native places, and other personal reasons [28–30]. In our study as well, the participants contemplated resigning their job, but had not done so out of a sense of duty because there was a huge demand for healthcare services and a shortage of HCPs. Among our participants, the sense of duty towards the patients outweighed not only their need for rest, but also their fears and need for appreciation and recognition. In some ways,

their sense of duty towards patients outweighed their sense of duty towards their family because, even though they feared getting infected and passing on the infection to their family, they had not stopped working as their patients needed them.

There has been inadequate infrastructural preparedness across the world [31] resulting in a shortage of oxygen cylinders, PPE, and essential drugs, leading to considerable negative psychological impact on HCPs [32]. The participants in this study were similarly negatively affected by the inadequate infrastructural preparedness in rural India. There was a perceived urban–rural disparity in infrastructural preparedness in India, leading to a negative psychological impact on the rural HCPs, reinforcing feelings of isolation and hopelessness as exhibited by our participants. Our study found that the participants strictly adhered to COVID-19 safety protocols. For a considerable duration of the pandemic, there were no vaccines available and a cure is still elusive. This may have contributed to the strict adherence to the safety protocols due to a sense of self-preservation. The uncertainty regarding the cure and lack of vaccines may have also contributed to the sense of hopelessness exhibited by the participants. This is in contrast to another study conducted in India that showed suboptimal adherence to COVID-19 safety guidelines among HCPs [33]. In our study, we did not evaluate the knowledge or practices of the HCPs with regard to the COVID-19 safety measures in detail. We only note that the HCPs said they were diligently following the guidelines. A detailed evaluation of the safety practices of the participants may very well yield results that agree with the previous study [33]; however, it is beyond the scope of this study.

A novel finding of our study was that the HCPs, who were putting their lives on the line, displayed zero tolerance towards persons who were flouting COVID-19 safety guidelines. Their belligerent reactions were magnified due to the increasing case count and they felt that their hard work and sacrifice was getting sabotaged by the careless violations of safety guidelines. Another novel finding of our study was that the HCPs perceived that the public at large regarded them to be profiteers and held them responsible for the high cost of beds and drugs to treat COVID-19 despite outwardly praising them as heroes, resulting in considerable mental distress to the HCPs.

A notable finding of our study is the fact that although the participants have delayed processing their emotions and some participants went so far as saying they were not able to cope with the mental challenges presented by the COVID-19 pandemic, they sought refuge in spirituality to help them get through their daily activities. Studies have shown that HCPs do not prefer availing help on mental health issues due to the stigma surrounding mental health in the healthcare community [34]. In a study of 11 doctors by Stanton & Randal [35], all participants, admitted to the existence of a culture in which they are expected to be a super-person and that their reputation in the medical community could be harmed if they showed any vulnerability. This stigma possibly arises from the belief that 'doctors are invincible' and 'illness is only for patients' [34]. Galbraith et al. [36] studied the attitudes of student nurses toward stress and help-seeking, and found out that only 1.4% of the participants preferred revealing their mental health issues, and the others cited reasons like stigma (24.7%), fear of damage to professional integrity (14.6%), and fear of facing career implications (45.2%) for not revealing their mental health issues. The study further expounded that the negative attitudes toward seeking help on mental health issues may have been inculcated even before training [36]. Spirituality is regarded as a powerful coping mechanism that can positively impact mental health [36], and given the stigma attached to mental health among HCPs [37], and the absence of stigma around spirituality, it is not surprising that HCPs preferred spirituality over traditional mental health support services or even peer support. Accordingly, we have devised a spiritual healing intervention to combat stress in HCPs and are running a clinical trial to assess the efficacy of this intervention.

A significant limitation of this study is that the lived experiences of the HCPs may vary depending upon their geographical locations and may not necessarily be similar to those of the HCPs in rural Dhanbad, India. Moreover, the age group of the participants ranges from 40 to 56 in this study. The lived experiences of the younger HCPs could differ from those of the older ones due to potential difference between their abilities to endure physical and mental strain, level of expertise, and expectations from themselves. Hence, these results cannot be generalized. Nevertheless, findings of this study can lay the foundation for future studies to investigate in detail the impact of COVID-19 on the professional and personal lives of HCPs, the difficulties they are facing in discharging their duties during the pandemic, and the mental health support they need. Another limitation of this study is that although our study yielded a significant and novel finding regarding the negative social perception of HCPs during the pandemic and it would be worth exploring the general views of society regarding HCPs before the pandemic and how this subsequently changed during the pandemic, we were unable to conduct such an analysis as there are no studies to conclusively establish the views of the public regarding the HCPs prior to the pandemic, and we failed to probe our participants about how they felt they were perceived prior to the pandemic. The impact of the COVID-19 pandemic on the social perception of HCPs is an interesting future avenue of research with important implications for the doctor–patient relationship.

Subsequent studies, each founded on the detailed investigation of case studies, like this one, can add to these findings, in order to establish more general claims. The emphasis on the individual making sense of their mental health related issues lays foundation for a rich dialogue with quantitative models in mental health psychology. Sense-making is fundamental to various models used by health psychologists to examine individuals' behaviours [6], and it is also central to how we have outlined the experience of the HCPs. Mixed method studies combining the findings of this experiential qualitative IPA study and that of the quantitative methods using standardized mental health screening tools such as the GAD-7, K-10, and DASS-21 can lead to a greater understanding of the HCPs' mental health status, which in turn can be used to devise stress management policies that are tailored for the HCPs.

This study demonstrates the shared themes of the participants while at the same time retains the individuals' experiences. By understanding a small number of individual cases in detail, we are in a better position to contemplate how, at a profound level, we share much in common with people whose circumstances may initially appear entirely different from our own. We hope that the readers can perceive the significance of individual accounts presented in this study in helping to comprehend what it is like to provide healthcare services during a pandemic in a resource-poor rural setting. Although the readers may not share the experience of these HCPs, we hope that they feel some sort of a resonance with the way it has impacted them existentially.

## 5. Conclusions

The ongoing COVID-19 pandemic has caused a significant amount of psychological distress to HCPs, despite which they prioritized their work, out of a sense of responsibility. Emotional responses such as fear, hopelessness, anger towards violators of COVID-19 safety guidelines etc. were common among most of the HCPs. They adopted different ways to cope with their day-to-day challenges, often resorting to spritualty. They gave a wide range of suggestions on how the situation can be improved—foremost of which was to increase the rural healthcare workforce. Taking their suggestions into consideration during policy formulation by the governments can help the HCPs. Designing psychological interventions to mitigate the psychological impact of the pandemic among HCPs around the theme of spirituality can be a future area of research.

## Supporting information

**S1 Appendix. Significant utterances of each participant and annotations of non-verbal cues that underpinned the interpretational phenomenological analysis.**
(DOCX)

## Acknowledgments

The authors would like to thank Ms. Santwana Sagnika, School of Computer Engineering, Kalinga Institute of Industrial Technology, Bhubaneswar, Odisha, India for administrative support, and Dr. Derek McPherson, Dunedin, Otago, New Zealand, for proofreading the revised manuscript.

## Author Contributions

**Conceptualization:** Laalithya Konduru.

**Data curation:** Laalithya Konduru, Nishant Das, Gargi Kothari-Speakman, Ajit Kumar Behura.

**Formal analysis:** Laalithya Konduru, Nishant Das, Ajit Kumar Behura.

**Funding acquisition:** Laalithya Konduru, Gargi Kothari-Speakman.

**Investigation:** Laalithya Konduru, Nishant Das.

**Methodology:** Laalithya Konduru, Gargi Kothari-Speakman.

**Resources:** Nishant Das.

**Supervision:** Laalithya Konduru.

**Validation:** Laalithya Konduru, Nishant Das, Gargi Kothari-Speakman, Ajit Kumar Behura.

**Visualization:** Gargi Kothari-Speakman.

**Writing – original draft:** Laalithya Konduru, Nishant Das.

**Writing – review & editing:** Laalithya Konduru, Gargi Kothari-Speakman, Ajit Kumar Behura.

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
