## [Decision Letter · Decision Letter 0]

3 Mar 2022

PONE-D-22-00877Experiencing the COVID-19 pandemic as a healthcare provider in rural Dhanbad, India: An interpretative phenomenological analysisPLOS ONE

Dear Dr. Konduru,

Thank you for submitting your manuscript to PLOS ONE. After careful consideration, we feel that it has merit but does not fully meet PLOS ONE’s publication criteria as it currently stands. Therefore, we invite you to submit a revised version of the manuscript that addresses the points raised during the review process.

Many thanks to the two reviewers for their excellent comments on this manuscript. I would be very grateful if you could consider all the points they raise in preparing a revised manuscript. From my own reading, I would also be grateful if you could provide a little information in the introduction on what has been found in other international locations, to provide some context on understanding the findings that you report.

We look forward to receiving your revised manuscript.

Kind regards,

Richard Rowe

Academic Editor

PLOS ONE

Journal Requirements:

2. Please note that in order to use the direct billing option the corresponding author must be affiliated with the chosen institute. Please either amend your manuscript to change the affiliation or corresponding author, or email us at plosone@plos.org with a request to remove this option."

Reviewers' comments:

Reviewer's Responses to Questions

**Comments to the Author**

1. Is the manuscript technically sound, and do the data support the conclusions?

Reviewer #1: Yes

Reviewer #2: Partly

2. Has the statistical analysis been performed appropriately and rigorously? 

Reviewer #1: N/A

Reviewer #2: N/A

3. Have the authors made all data underlying the findings in their manuscript fully available?

Reviewer #1: Yes

Reviewer #2: No

4. Is the manuscript presented in an intelligible fashion and written in standard English?

Reviewer #1: Yes

Reviewer #2: Yes

5. Review Comments to the Author

Reviewer #1: General Points

This is an excellent and very interesting paper which provides a unique perspective regarding the experiences of healthcare workers in a rural setting – contrasting much of the current literature which focuses upon larger urban / teaching hospitals. It highlights the challenges associated with facing the COVID-19 pandemic in a severely under-resourced setting, which will be useful for the provision of support in future similar crises.

Have suggested minor clarifications – particularly for readers who are not familiar with the culture or rural healthcare environment.

Abstract

- Suggest correcting second sentence to: ‘it has led to a myriad of health problems’

- Clarification of sampling method (see longer comment under methods heading)

- Suggest correcting to “participants were under mental duress” or “suffered from mental duress”

- Could highlight uniqueness of findings and study setting more in the abstract

Introduction

- Could consider elaborating on the challenges faced in the rural setting for readers not familiar with this environment

Methods

- Please clarify what you mean by purposive heterogenous snowball sampling – your explanation in the manuscript sounds like purposive sampling rather than snowball sampling (where initial participants identify further suitable participants).

- How did you identify the 5 participants?

- Could expand on how you defined data saturation – did this occur as interviews were ongoing or as a review process after 5 participants were recruited?

Results

- I think the results of this study are very rich and tell a fascinating story, providing novel insights into pandemic management in a rural setting.

- 3.2.1 c. This is a very interesting data subtheme – would it be worth expanding about general views of society regarding HCW prior to the pandemic? Did this change?

- Can you clarify what you mean by ‘stepmotherly treatment by the government’?

- Could you expand / reference examples of stigma regarding mental health issues in the healthcare community?

Discussion

- Consider providing more specific suggestions for future change based upon the detailed and novel findings of your study

Reviewer #2: This is a very timely contribution to the body of literature related to healthcare workers’ perspectives during Covid-19. However below are some concerns which authors can address before this manuscript is considered for publication:

1. The authors used interpretative phenomenological analysis (IPA) but it lacks some link with the theoretical underpinnings of IPA. For instance, IPA is an idiographic approach, hence, the authors need to present the results by illuminating an individual healthcare worker’s perspective in addition to the group themes currently presented, in order to alien the result with IPA’s individualistic/idiographic approach. It is highly recommended to consult the latest/revised terminologies of results/themes in an IPA study i.e., Personal experiential themes and group experiential themes. Full details of the new terminology and worked examples of their use in practice are provided in the new books:

Smith JA , & Nizza I (2021) Essentials of Interpretative Phenomenological Analysis. Washington DC: APA.

Smith JA, Flowers P, Larkin M (2022) Interpretative Phenomenological Analysis: Theory, Method, Research. London: Sage by Smith

However, if the authors do not want to use the new terminologies, still it is recommended to highlight at least the idiographic focus in the result section.

2. Furthermore, the authors need to also inculcate the hermeneutic and double hermeneutic stance of IPA in this study by not just descriptively presenting their own opinion after each theme but go deeper in the analysis and present descriptive, linguistic and conceptual analysis of each theme.

3. The following statement in the result section of the abstract needs to be rephrased to bring more clarity “Our findings demonstrate that the participants were mental duress due to heavy workloads”.

4. Also, start the result section in the abstract by presenting the total number of super-ordinate and subordinate themes and their labels.

5. In the method section, need to elaborate more how IPA is the best suited method in this study with more citations.

6. Furthermore, authors need to explain within their data collection section (with citations) why unstructured interviews were used whereas in IPA the semi-structured interviews are highly recommended.

7. Add some research, policy, and clinical implications in the discussion section

6. PLOS authors have the option to publish the peer review history of their article (what does this mean?). If published, this will include your full peer review and any attached files.

Reviewer #1: **Yes: **Kate Grailey

Reviewer #2: **Yes: **Dr Fahad Riaz Choudhry

---

## [Author Response · Author response to Decision Letter 0]

20 Apr 2022

Point-by-point responses to the reviewers has been attached

---

## [Decision Letter · Decision Letter 1]

8 Aug 2022

PONE-D-22-00877R1Experiencing the COVID-19 pandemic as a healthcare provider in rural Dhanbad, India: An interpretative phenomenological analysisPLOS ONE

Dear Dr. Konduru,

Thank you for submitting your manuscript to PLOS ONE. After careful consideration, we feel that it has merit but does not fully meet PLOS ONE’s publication criteria as it currently stands. Therefore, we invite you to submit a revised version of the manuscript that addresses the points raised during the review process.

Very many thanks for attending to the comments of the reviewers so carefully and for your patience with the review process. As you will see below I have received comments back from Reviewer 1 who was satisfied with the revisions.  I have also read the manuscript carefully again myself and had the following thoughts. Once you have addressed these minor issues then I expect to be able to make a final decision on the submission. 

Line 46. Update COVID stats to be more recent

Line 66 To date not Till date

Line 87. Ensure the meaning of the IPA abbreviation is spelt out on first mention, not subsequent mentions (currently line 96).

Line 90 Aims to, not attempts to

Line 178. Varun is described as male and having a husband. As no other mention of him being in a same-sex relationship was evident in my reading, I wonder if “marital partner” might be a better term to use here, to save the reader from wondering whether a same-sex relationship was relevant to the experience of the pandemic.

Line 194 Is history of diabetes appropriate? In my lay understanding is a chronic condition that is managed, so wouldn’t this participant still have diabetes? I would have thought “diabetic status” is appropriate, but please use whatever description you think is most accurate.

Line 226-228 Sentence beginning “This may be because…” seems to be very speculative to me, as it is based on what the participant does not say rather than what they said. Please consider deleting it. 

Table 2, theme 1. Please provide an alternative term to “loot” as I am not sure what that would mean in this context. Having read further, I think “financial exploitation” might be clearer. Or perhaps profiteering.

Table 2, theme 3. The justification on the theme name seems odd, please re-space.

Line 384 Define aerosolization for a non-specialist audience

Line 402 Some reason or other (delete “the”)

Line 528 Off from not Off off

Line 586 Highlighted the (delete “of”)

Line 594-664 This paragraph is too long and needs to be broken up.

Line 659 help is repeated.

Line 693-695 This paragraph is not sufficiently developed as indicated by it forming only a single sentence. I recommend developing it or removing it.

Line 705 has caused A significant amount… (insert A)

We look forward to receiving your revised manuscript.

Kind regards,

Richard Rowe

Academic Editor

PLOS ONE

Journal Requirements:

Reviewers' comments:

Reviewer's Responses to Questions

**Comments to the Author**

1. If the authors have adequately addressed your comments raised in a previous round of review and you feel that this manuscript is now acceptable for publication, you may indicate that here to bypass the “Comments to the Author” section, enter your conflict of interest statement in the “Confidential to Editor” section, and submit your "Accept" recommendation.

Reviewer #1: All comments have been addressed

2. Is the manuscript technically sound, and do the data support the conclusions?

Reviewer #1: (No Response)

3. Has the statistical analysis been performed appropriately and rigorously? 

Reviewer #1: (No Response)

4. Have the authors made all data underlying the findings in their manuscript fully available?

Reviewer #1: (No Response)

5. Is the manuscript presented in an intelligible fashion and written in standard English?

Reviewer #1: (No Response)

6. Review Comments to the Author

Reviewer #1: (No Response)

7. PLOS authors have the option to publish the peer review history of their article (what does this mean?). If published, this will include your full peer review and any attached files.

Reviewer #1: **Yes: **K E Grailey

---

## [Author Response · Author response to Decision Letter 1]

10 Aug 2022

Response in file named response to reviewers

---

## [Editor Report · Decision Letter 2]

11 Aug 2022

Experiencing the COVID-19 pandemic as a healthcare provider in rural Dhanbad, India: An interpretative phenomenological analysis

PONE-D-22-00877R2

Dear Dr. Konduru,

We’re pleased to inform you that your manuscript has been judged scientifically suitable for publication and will be formally accepted for publication once it meets all outstanding technical requirements.

Kind regards,

Richard Rowe

Academic Editor

PLOS ONE
---

## [Editor Report · Acceptance letter]

16 Aug 2022

PONE-D-22-00877R2 

Experiencing the COVID-19 pandemic as a healthcare provider in rural Dhanbad, India: An interpretative phenomenological analysis 

Dear Dr. Konduru:

I'm pleased to inform you that your manuscript has been deemed suitable for publication in PLOS ONE. Congratulations! Your manuscript is now with our production department. 

Kind regards, 

on behalf of

Professor Richard Rowe 

Academic Editor

PLOS ONE